# Modified *Camellia oleifera* Shell Carbon with Enhanced Performance for the Adsorption of Cooking Fumes

**DOI:** 10.3390/nano11051349

**Published:** 2021-05-20

**Authors:** Dongliang Liao, Wen Shi, Jing Gao, Bin Deng, Ruijin Yu

**Affiliations:** 1College of Natural Resources and Environment, Northwest A&F University, Yangling 712100, China; ldl19990923@163.com; 2College of Chemistry & Pharmacy, Northwest A&F University, Yangling 712100, China; wznln@nwafu.edu.cn (W.S.); Sven@nwafu.edu.cn (J.G.); 3College of Chemistry & Biology and Environmental Engineering, Xiangnan University, Chenzhou 423043, China; 4Hunan Provincial Key Laboratory of Xiangnan Rare-Precious Metals Compounds Research and Application, Chenzhou 423043, China

**Keywords:** cooking fumes, modification, carbon from *Camellia oleifera* shell, adsorption performance

## Abstract

Using *Camellia oleifera* shell (COS) as a raw material and phosphoric acid as the activator, activated *Camellia oleifera* shell carbon (COSC-0) was prepared and then modified by Fenton’s reagent (named as COSC-1). SEM, GC-MS, FTIR, and specific surface area and pore analyzers were used to study the adsorption performance of COS, COSC-0, and COSC-1 on cooking fumes. Results showed that COSC-1 was the best adsorbent compared with COS and COSC-0. The adsorption quantity and penetrating time of COSC-1 were 44.04 mg/g and 4.1 h, respectively. Most aldehydes could be adsorbed by COSC-1, which was due to the large number of carbonyl and carboxyl groups generated on the surface of COSC-1 from the action of Fenton’s reagent. The adsorption effect of COSC-1 on different types of pollutants in cooking fumes was analyzed based on the similar compatibility principle. COSC-1 showed a much higher adsorption effect on the strong polarity functional groups than on weak polar groups. The results provide a theoretical basis for the application of *Camellia oleifera* shell carbon adsorption technology in the treatment of cooking fumes.

## 1. Introduction

Cooking fumes from the Chinese catering industry contain numerous carcinogenic substances such as aldehydes and polycyclic aromatic hydrocarbons (PAHs) [1,2,3]. Recently, research on female non-smokers showed that people tend to suffer lung cancer due to the polymorphism of human genes and the interaction of fumes [4,5,6,7].

In the last decade, biological washing and catalyst combustion have been successfully applied in the field of cooking fume purification [8,9], but complicated operating conditions and expensive cost are their main disadvantages. Thus, a simple and effective method to degrade and treat cooking fumes has become an urgent need. Activated carbon (AC) is widely applied in dealing with waste gas due to its unique properties, such as good pore structures and high specific surface areas [10,11]. Recently, AC has become a popular material because many solid wastes such as straws [12,13], nut shells [14,15,16,17], almond shells [18,19,20,21], and bagasse [22,23] can be utilized to prepare AC [1].

*Camellia oleifera* shell (COS) is an important byproduct in processing woody edible oil. Approximately 0.54 tons of COS can be produced from 1 ton of *Camellia oleifera* nuts, and their main components are cellulose, hemicellulose, and lignin; thus, COS is an ideal source for the preparation of AC [24]. In recent years, many researchers have successfully prepared AC using COS and explored its adsorption performance by using different preparation methods. For instance, Sun et al. [25] used phosphoric acid as an activator and microwave heat to prepare AC from the oil of camellia nut shells, and the adsorption for methylene blue reached 330 mg/g. Zheng et al. [26] prepared an adsorbent by pyrolysis carbonization followed by the potassium hydroxide (KOH) activation of COS under the nitrogen atmosphere; this C. oleifera shell activated carbon adsorbent can effectively adsorb hexavalent chromium (Cr(VI)) from aqueous solution. Zirconium dioxide-biochar (ZrO_2_/BC), with excellent fluoride adsorption properties, was also successfully prepared by calcining the zirconium-impregnated byproduct from *Camellia oleifera* (C. oleifera) seed shell in a one-step method [27]. Therefore, *Camellia oleifera* shell carbon (COSC) has attracted increasing attention as a promising material in chemical engineering and environmental science because of its large surface area, large pore size, and high stability. However, studies on COSC for the adsorption of cooking fumes are limited. In this research, an AC from COS modified with Fenton’s reagent for the treatment of cooking fumes was reported for the first time.

### 1.1. Preparation and Modification of COSC

In a typical experimental procedure, COS taken from the countryside of Ningdu, Jiangxi Province of China was first cleaned with water and dried in an oven at 12 °C for 12 h. Then, the dried COS was added into 60% of phosphoric acid solution at a weight ratio of 1:3. The mixture was placed into a box-type resistance furnace for carbonization and activation at 500 °C for 2 h with a heating rate of 10 °C min^−1^. When it cooled down to room temperature, the product was collected and washed with deionized water until the pH of the solution was close to neutral. Finally, the product was dried and stored in a sample bag named COSC-0.

Fenton’s reagent was prepared from a solution of FeSO_4_·7H_2_O (0.5 mol/L) and H_2_O_2_ (30%) with the molar proportion of n(Fe): n(H_2_O_2_) = 1:10, and the pH value of the solution was adjusted to 3 with 0.1 mol/L H_2_SO_4_ solution. The COSC-0 was immersed in Fenton’s reagent and left overnight at room temperature. The obtained product was washed with deionized water to neutral and dried at 50 °C in the oven, and the corresponding samples were denoted as “COSC-1.”

### 1.2. Concentration Measurement of Pollutants from Cooking Fumes

As shown in Figure 1, the samples of simulated cooking fumes (SCFs) were collected for 3 min in the inlet and outlet with sampling tubes filled with 5 g of granular AC; sampling was repeated three times. The granular AC in the sample tube was removed, transferred into a flask, and washed with 15 mL of carbon tetrachloride for 5 min by using an ultrasonic device (Hangzhou Boke Ultrasonic Equipment Co., LTD, Hangzhou, Zhejiang, China). The cleaning liquid was collected and transferred into another flask (marked as “A”). The granular AC was washed once more with 10 mL of carbon tetrachloride, and the cleaning liquid was also collected and transferred into flask A.

According to the standard curve, the pollutant concentration of SCFs in the collected cleaning liquid (ρ_0_) was measured with a CY-2000 multi-functional IR oil content analyzer. The pollutant concentration of SCF samples (*C*_0_) could be calculated by the following equation:*C*_0_ = (60 × ρ_0_ × V) ÷ (Q × t)(1)
where *C*_0_ is the pollutant concentration of the sample, mg/m^3^; ρ_0_ is the pollutant concentration of SCFs in the collected cleaning liquid, mg/L; V is the volume of cleaning liquid, L; Q is the gas flow, m^3^/h; and t is the sampling time, min.

### 1.3. Measurement of Adsorption Capacity

To measure the adsorption capacity of different adsorbents for cooking fumes, a fixed adsorbing tower was set up and filled with a 30 cm thick adsorbent. The inlet concentration of SCFs was 90 mg/m^3^, the gas hour space velocity (GHSV) was 4300/h, the temperature was 25 ± 5 °C, and the outlet samples of SCFs were collected at different times. Breakthrough curves of different adsorbents for SCFs were drawn with adsorption time as the abscissa and outlet concentration of SCFs as the ordinate. The outlet concentration of 2.0 mg/m^3^ acted as the penetration concentration of SCFs, based on GB18483-2001 in China, and the corresponding adsorption time and adsorption quantity are described as penetration time and penetration adsorption quantity. Moreover, the saturation act time was the adsorption time when the proportion of outlet and inlet concentrations of cooking fumes was 95%. Through graphing software of Origin 9.0 (*Origin* Lab), the penetration adsorption quantity of cooking fumes could be calculated from the integral area (S) between the penetration curves and abscissa:*q* = (*C* × *t* − *S*) × *µ* × *A* × 3600 ÷ *M*(2)
where *q* is the equilibrium adsorption quantity of adsorbents for cooking fumes, mg/g; *C* is the inlet concentration of cooking fumes, mg/m^3^; *t* is the adsorption time, h; *µ* is the velocity of the bed layers, m/s; *A* is the sectional area of adsorption tower, m^2^; and *M* is the weight of adsorbents, g.

### 1.4. GC-MS Analysis of Organic Contents in Cooking Fumes

GC conditions: The chromatographic column was an Elite-5MS capillary column (30 m × 0.25 mm, 0.25 µm). The carrier gas was helium with a flow rate of 1 mL/min. The collision gas was argon with a temperature of 250 °C, flowing amount of 1 µL, and division ratio of 10:1. The column box was maintained at 50 °C for 2 min, programmed at 8 °C/min to 250 °C, and held at 250 °C for 12 min.

MS conditions: The injector temperature was 250 °C. The electric impact energy of EI ion source was 70 eV. The mass scanning range was 40–550 *m/z*. The delaying time for solvent was 3 min.

### 1.5. Characterization

The morphology of the samples was investigated by scanning electron microscopy (SEM) (S-4800, Hitachi, Japan). FTIR was used to characterize and analyze the framework and functional groups of samples. The scanning range was from 400 cm^−1^ to 4000 cm^−1^. The specific surface area and pore structure of samples were determined on a NOVA1200e Surface Area Analyzer (NOVA1200e, Quantachrome, America) at 77.4 K. All samples were heated at 120 °C for 14 h for degassing treatment before measurement. The specific surface area and pore size distribution of the samples were obtained from the N_2_ adsorption desorption isotherm by the Brunauer–Emmett–Teller (BET) method and the quenched solid density functional theory (QSDFT) method [28], respectively. The total pore volume was obtained at a relative pressure of *P/P*_0_ = 0.99.

## 2. Results and Analysis

Figure 2 shows the adsorption breakthrough curves of COS, COSC-0, and COSC-1 for cooking fumes. The penetration times of COS, COSC-0, and COSC-1 were 0.9, 2.1, and 4.1 h, respectively, which indicated that the effective action time of COSC-1 was the longest among the tested materials. Figure 3 illustrates the adsorption quantity of COS, COSC-0, and COSC-1 for cooking fumes. The adsorption quantities of COS, COSC-0, and COSC-1 for cooking fumes were 6.43, 22.58, and 44.04 mg/g, respectively, which showed that the adsorption quantity of COSC-1 was the largest. Thus, COSC-1 demonstrated better adsorption properties than COS and COSC-0.

GC-MS analysis was conducted to identify the adsorption component of cooking fumes by COS, COSC-0, and COSC-1. The detected species and their relative abundances are listed in Table 1 and Figure 4. The results showed that the main compositions of cooking fumes were aldehydes and substituent olefins. Cheng et al. [29] carried out a systematic study on aldehydes and ketones in the exhausts of eight Beijing restaurants, and found that the concentrations of aldehydes and ketones (C1–C9) were in the range of 115.47–1035.99 μg.m^−3^ and the percentages of C1–C3 were above 40%. Compared with COS and COSC-0, COSC-1 demonstrated the best adsorption capacity for all the components of cooking fumes, as shown in Table 1, which was due to the dominant chemical adsorption performance resulting from the strong influence of the Fenton’s reagent. However, COS, COSC-0, and COSC-1 exhibited a similar and better adsorption capacity for aldehydes than that for substituted olefins, such as p-propenyl phenyl methyl ether, N-benzyl-allyl amine, 1,5-diphenyl-3-(2-ethyl benzene)-2-amylene, timnodonic acid, and 2-methyl-6-benzene-1,6-heptyl diene. This is because aldehyde compounds are a class of volatile organic compounds with strong chemical reactivity, which were easily adsorbed by COS, COSC-0, and COSC-1.

To further investigate the adsorption behavior of COS, COSC-0, and COSC-1, the morphology and structure of COS, COSC-0, and COSC-1 were characterized by specific surface area and pore analyzers, SEM, and FTIR.

N_2_ of the adsorption–desorption isotherms obtained at 77.4 K of COS, COSC-0, and COSC-1 is shown in Figure 5. The figure shows that the isotherms of COSC-0 and COSC-1 belonged to capillary condensation (IV) according to the definition of IUPAC [30]. The adsorption capacity increased with the increase in relative pressure, and the adsorption volume of N_2_ sharply ascended when the relative pressure reached 1.0, which indicated the existence of mesoporous and macroporous pores in those adsorbents [31]. Figure 6 shows the pore size distribution of COSC-0 and COSC-1. The pore size of COSC-0 and COSC-1 exhibited centralized distribution, and the pore apertures were mainly distributed in the range of 3.2–4.5 nm, which confirmed the mesoporous structure of COSC-0 and COSC-1. It has been confirmed that components with a large molecular diameter are always more easily absorbed by adsorbents with large apertures than their counterparts. The components of aldehydes and substituent olefins from cooking fumes have a macromolecular structure, which means that they are easily absorbed by COSC-0 and COSC-1 with mesoporous structures.

Table 2 shows the specific surface areas and porous structures of COSC-0 and COSC-1. Compared with COSC-0, COSC-1 showed a decrease in the specific surface areas and pore volumes and a slight increase in aperture, because part of the porous microstructure of COSC was collapsed by the oxidation of H_2_O_2_ after treatment with Fenton’s reagent. Moreno-Castilla et al. [32] and Bandosz Teresa et al. [33] also found that H_2_O_2_ modification can result in the reduction in the specific surface area of AC to different degrees. Figure 7 shows the SEM images of COS, COSC-0, and COSC-1. Large quantities of substances were adsorbed on the surface or filled in the pores of COSC-1 (Figure 7(c1,c2)), which proved the decrease in specific surface areas and pore volumes of COSC-1.

Figure 8 shows the FTIR spectra of COSC-0 and COSC-1. The broad and strong absorption peak at 3417 cm^−1^ was related to the stretching vibration of -OH. The absorption peak at 1620 cm^−1^ was the flexural vibration peak of hydroxyl (the physical absorbed water), which indicated that some water molecules were introduced into the surface and the pores of adsorbent in the form of absorbed water. The absorption bands, which were formed by skeletal vibrations, were observed at 1714, 1620, 1384, 1036, 875, and 592 cm^−1^. The peak at 1714 cm^−1^ corresponded to the stretching vibration peak of C=O in the carbonyl and carboxyl [34]. The steep peak at 1384 cm^−1^ was the stretching vibration peak of -NO. The broad and strong vibrations at 1036 and 1116 cm^−1^ were the symmetrical stretching vibrations of S=O bonds in SO_4_^2−^. The peak at 875 cm^−1^ corresponded to the symmetrical vibration [35] of P-O-P in polyphosphate. The peak at 592 cm^−1^ belonged to the lattice vibration of cations (Fe^2+^, Fe^3+^, Al^3+^, and Mg^2+^) [36], whose location was the same as the stretching vibration of Fe-O. As shown in Figure 8, the new peak at 1714 cm^−1^ appeared and the peak at 592 cm^−1^ became apparent in the sample of COSC-1, indicating that a large number of carbonyl and carboxyl groups and Fe ions generated after COSC-0 were modified by Fenton’s reagent. Compared with COSC-0, the absorption peaks of the carbonyl and carboxyl groups in COSC-1 were stronger and broader. Thus, COSC-1 demonstrated the best absorption performance for organic contaminants in cooking fumes among the tested materials.

The large number of carbonyl and carboxyl groups generated on the surface of COSC-1 resulted from the action of Fenton’s reagent. The catalytic effect of Fe^2+^ and Fe^3+^ formed in the Fenton system was favored in the conversion of H_2_O_2_ into HO· and HOO·, as shown in Equations (3) and (4). HO· radicals have strong addition reaction and oxidation capacity due to their electron-deficient group and high electric potential (+2.8 V) [11,37,38]. Therefore, radicals of HO· or HOO· could attack unsaturated double bonds and defects on the surface of COSC-1 and addition reactions could occur, which resulted in the carbonyl and carboxyl functional groups forming on the surface of COSC-1. In addition, HO· can further oxidize unstable –CH_2_OH and –CHOH– functional groups into carboxyl groups on the surface of COSC-1 [39,40]. Thus, the surface modification of COSC-0 was realized, and COSC-1 with a large number of carbonyl and carboxyl groups was formed.
Fe^2+^ + H_2_O_2_ → Fe^3+^ + HO·+ OH^−^(3)
Fe^3+^ + H_2_O_2_ → Fe^2+^ + HOO·+ H^+^(4)

Compared with COSC-0, COSC-1 presented better adsorption capacity because it is rich in carbonyl and carboxyl groups, which can react with carboxylic acids and alcohols from cooking fumes to improve the adsorption capacity of COSC-1. Moreover, during the preparation of COSC-1, some substances, such as HO, Fe^3+^, Fe^2+^, and H_2_O_2_, may remain on the surface or the inner wall of the pore of COSC-1. These substances can oxidize and decompose most of the oil fume pollutants adsorbed on the surface of COSC-1, thereby increasing the adsorption capacity of COSC-1.

Table 1 shows that all of the adsorption capacities of COS, COSC-0, and COSC-1 for aldehydes were 100%, which indicated that aldehydes could easily be adsorbed by COS, COSC-0, and COSC-1. According to the similarity principle that similar substance is more likely to be dissolved by each other, aldehydes were easily compatible with carbonyl and carboxyl groups on the surface of COS, COSC-0 and COSC-1 due to their similar polarities [41,42]. By contrast, the polarity of functional groups (–C=C–of olefins was weaker than that of the functional groups on the surface of COS, COSC-0 and COSC-1. Therefore, the adsorption performance of COS, COSC-0, and COSC-1 on olefins was weaker than that on aldehydes.

Recently, Yu et al. [43] reported a novel ferrisilicate MEL zeolite applied for the removal of non-methane hydrocarbon (NMHC) from cooking oil fumes (COFs). A comparison of the performance on the adsorption of cooking fumes between COSC with ferrisilicate MEL zeolites was made, as shown in Table 3. COSC-1 presented the best adsorption performance for cooking fumes, which indicates that COSC-1 is a good material for cooking fume purification.

## 3. Conclusions

A cheap adsorbent (COSC-1), which was activated by phosphoric acid and then modified by Fenton’s reagent from COSC, has been successfully prepared, and the adsorption properties for cooking fume were studied. Results showed that COSC-1 was the best adsorbent compared with COS and COSC-0. The superior adsorption properties of COSC-1 were due to the large number of carbonyl and carboxyl groups generated on the surface of COSC-1 from the action of Fenton’s reagent. The adsorption effects of COSC-1 on different types of pollutants of cooking fume were analyzed based on the similar compatibility principle. The adsorption effect of COSC-1 on strong polarity functional groups was much higher than that on weak polar groups. COSC-1 also exhibited a far better performance on the adsorption of cooking fumes compared with ferrisilicate MEL zeolites. This work provides a theoretical basis for the application of COSC adsorption technology in the treatment of cooking fumes.

## Figures and Tables

**Figure 1 nanomaterials-11-01349-f001:**
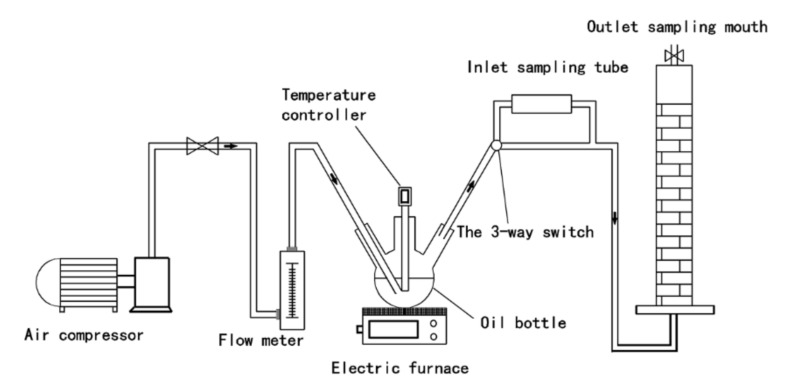
Schematic of experimental apparatus for treating cooking fumes by COS, COSC-0, and COSC-1.

**Figure 2 nanomaterials-11-01349-f002:**
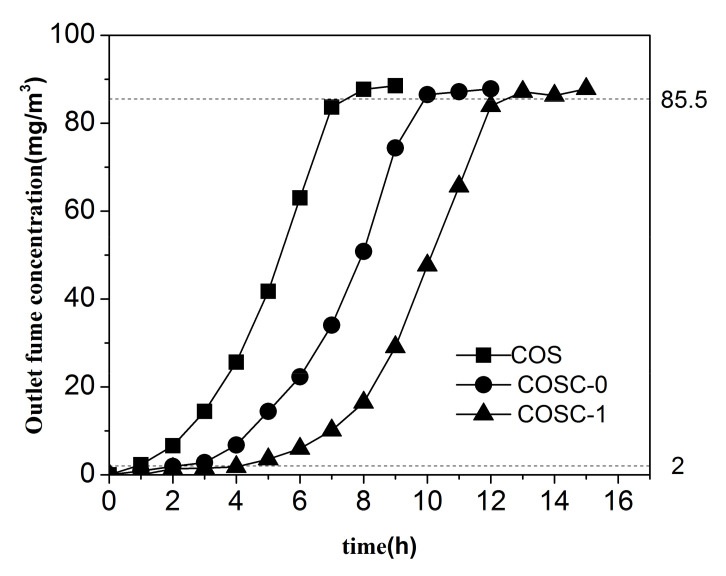
Adsorption breakthrough curves of different adsorbents for cooking fumes (*GHSV*: 4300/h, *T* = 25 °C, *C*_0_ = 90 mg/m^3^, *H* = 30 cm).

**Figure 3 nanomaterials-11-01349-f003:**
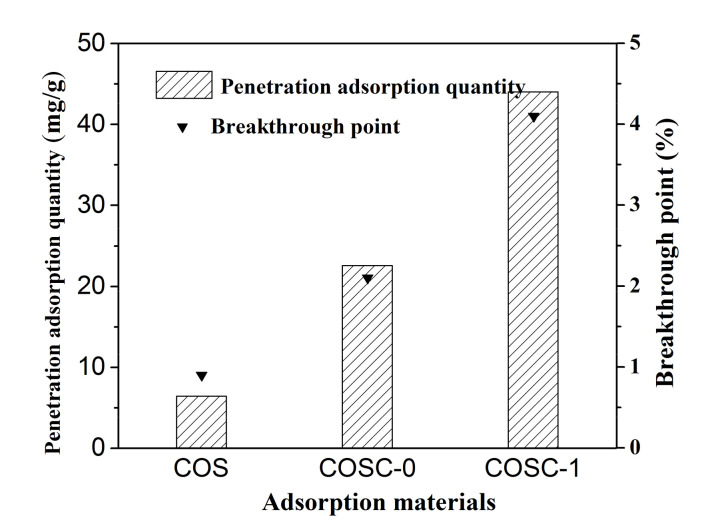
Adsorption quantity of different adsorbents for cooking fumes (*GHSV*: 4300/h, *T* = 25 °C, *C*_0_ = 90 mg/m^3^, *H* = 30 cm).

**Figure 4 nanomaterials-11-01349-f004:**
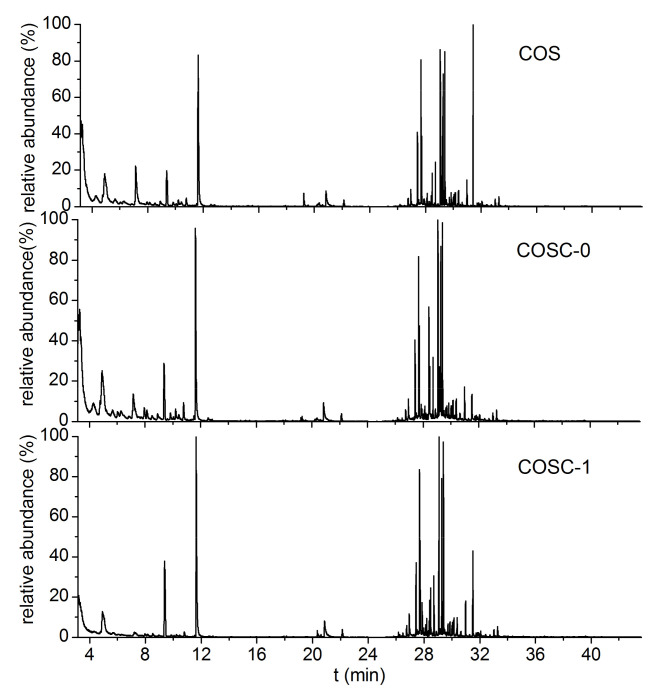
Total ionic chromatogram of absorbent components after treating cooking fumes.

**Figure 5 nanomaterials-11-01349-f005:**
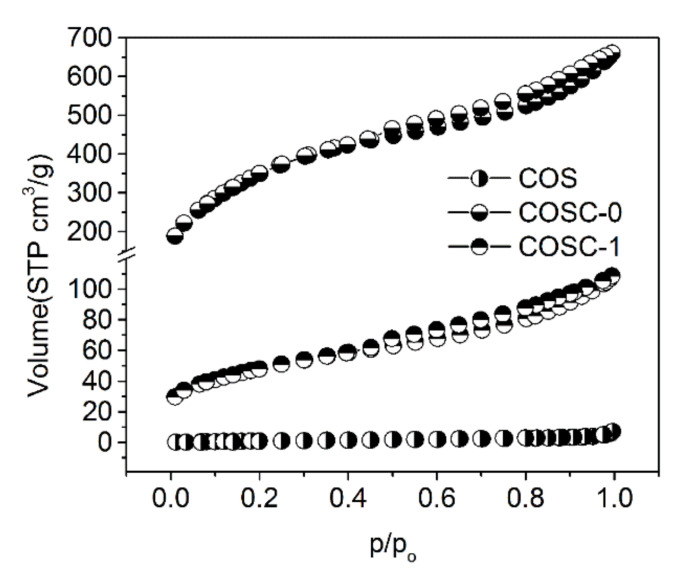
Nitrogen adsorption–desorption isotherm of COS, COSC-0, and COSC-1.

**Figure 6 nanomaterials-11-01349-f006:**
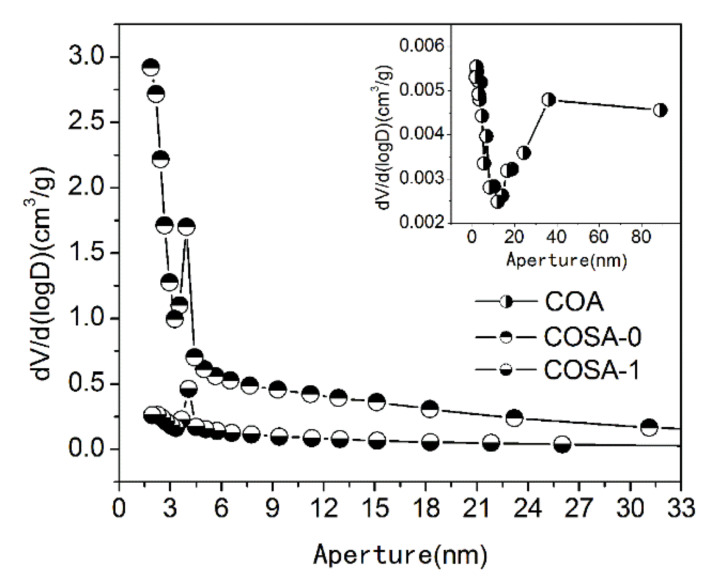
Pore size distribution of COS, COSC-0, and COSC-1.

**Figure 7 nanomaterials-11-01349-f007:**
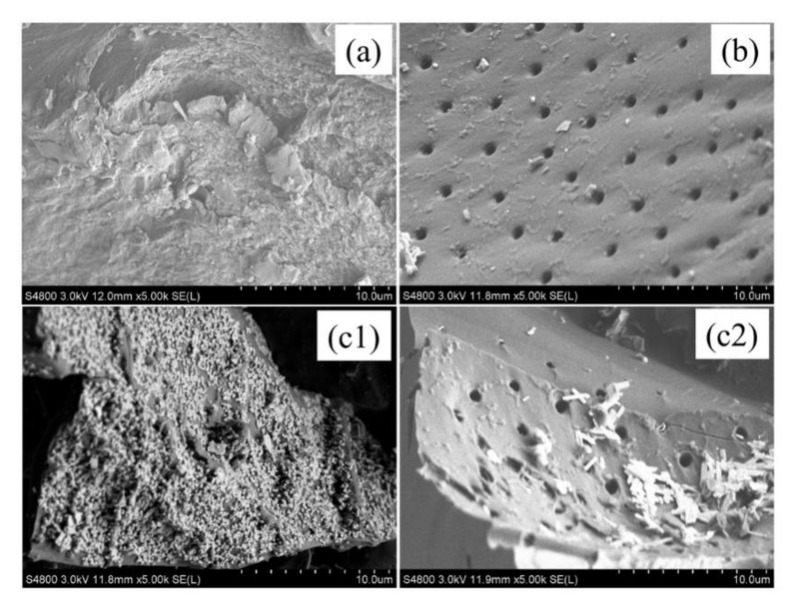
SEM images of the three kinds of materials. COS: (**a**); COSC-0: (**b**); COSC-1: (**c1**), (**c2**).

**Figure 8 nanomaterials-11-01349-f008:**
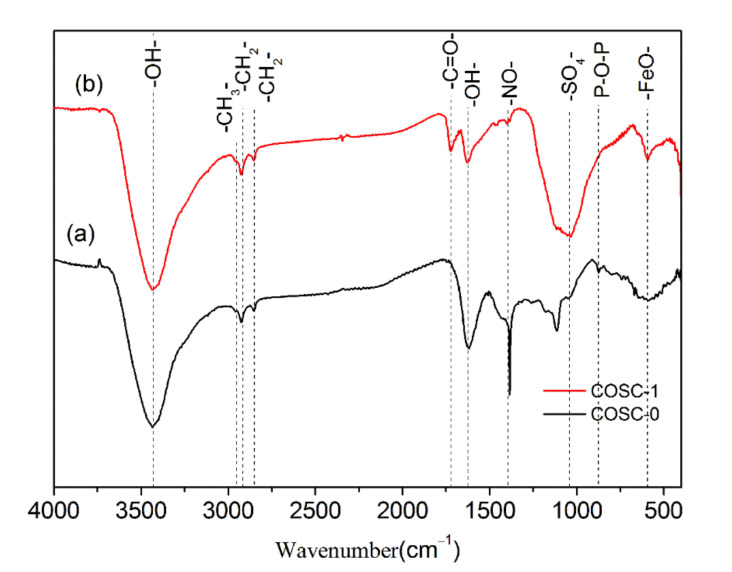
FTIR spectrum of COSC-0 (**a**) and COSC-1 (**b**).

**Table 1 nanomaterials-11-01349-t001:** Results of the cooking fume pollutants treated by different absorbents.

No.	t (min)	Name of Chemical Compound	Molecular Formula	Removal Rate (%)		
				COS	COSC-0	COSC-1
1	3.189	2, 7-dimethyl-1-octanol	C_10_H_22_O	100	100	100
2	4.965	n-heptaldehyde	C_7_H_14_O	100	100	100
3	6.263	(2Z)-Heptenal	C_7_H_12_O	100	100	100
4	7.051	2-Pentylfuran	C_9_H_14_O	40	67	100
5	7.421	Octanal	C_8_H_16_O	100	100	100
6	8.949	(E)-2-Octena	C_8_H_14_O	100	100	100
7	10.7	1-hexyl-1-cyclopentene	C_11_H_20_	67	48	100
8	11.58	p-propenyl phenyl methyl ether	C_10_H_12_O	27	37	46
9	11.8	trans-2-nonenal	C_9_H_16_O	100	100	100
10	14.58	trans-2-decyl olefine aldehyde	C_10_H_18_O	100	100	100
11	15.35	2-octyl-tetrahydro-furan	C_12_H_20_O	100	100	100
12	16.11	2, 4-decadienal	C_10_H_16_O	100	100	100
13	17.27	2-undecenal	C_11_H_20_O	100	100	100
14	20.78	N-benzal-allyl amine	C_10_H_11_N	31	46	53
15	26.92	1, 2-diphenyl cyclopropane	C_15_H_14_	22	31	43
16	27.93	linalyl isobutyrate	C_14_H_24_O_2_	100	100	100
17	28.72	3-DNA-estradiol	C_18_H_24_O	27	28	44
18	29.08	1, 5-diphenyl-3-(2-ethyl benzene)-2-amylene	C_25_H_26_	30	40	50
19	29.42	timnodonic acid	C_20_H_30_O_2_	22	33	45
20	30.41	5, 7-dodecane acetylene 2-1, 12-dio	C_12_H_18_O_2_	29	36	52
21	31.02	[(2, 3-diphenyl propyl) methyl]-phenyl sulfur	C_22_H_20_OS	26	100	100
22	33.34	2-methyl-6-benzene-1, 6-heptyl diene	C_14_H_18_	41	52	64

Annotation: 1–22 are the tested pollutants during the process of cooking fume suction.

**Table 2 nanomaterials-11-01349-t002:** Specific surface area and pore structures of COSC-0 and COSC-1.

Samples	COSC-0	COSC-1
*S_BET_*/(m^2^/g)	1245	934
*V_total_*/(cm^3^/g)	1.02	0.69
*D_p_*/(nm)	3.284	4.021

**Table 3 nanomaterials-11-01349-t003:** Adsorption performance of different adsorbents for cooking fumes.

Adsorbent	*S_BET_*/(m^2^/g)	*V_total_*/(cm^3^/g)	*PAQ*/(mg/g)	References
COSC-0	1244.7	1.02	6.43	This study
COSC-1	933.5	0.69	22.58	This study
Sam-SiFe (II)	388.0	0.23	4.468	[43]
Sam-SiFe (III)	411.4	0.23	3.659	[43]
Sam-SiAl	403.6	0.23	2.781	[43]

PAQ: penetration adsorption quantities. Sam-SiFe (II), Sam-SiFe (III): Fe-containing zeolites. Sam-SiAl: Al-containing zeolites.

## Data Availability

Data is contained within the article.

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
