# Peer review of "Modified Camellia oleifera Shell Carbon with Enhanced Performance for the Adsorption of Cooking Fumes"

_nanomaterials, 2021, doi:10.3390/nano11051349_

Round 1

Reviewer 1 Report

This paper studies the adsorption capacity of a biomass-based activated carbon for cooking fumes. The biomass in question is not extensively reported in the literature. The authors obtained a good performance for these materials.

Nevertheless, the paper needs English revision, because some sentences are not clear. All the sections could be improved.

In my perspective, the following points must be addressed for the improvement of the manuscript and further publication:

  1. I would advise to reformulate the title. I would suggest something like: “Modified camellia oleifera shell carbon with enhanced performance for the adsorption of cooking fumes”, or similar. It would be more informative and clearer.
  2. Lines 39-41: “However, AC has become a kind of scarce resource because some solid wastes such as straws [12, 13], nut shells [14-17], almond shells [18-21], and bagasse [22, 23] are increasingly becoming utilized to prepare AC [1]”. I don’t quite understand this sentence, if the biomasses are increasingly being used, AC would not be a scarce resource…
  3. Line 54. Please, replace “aslo” for “also”.
  4. Lines 60-61. “In this research, an AC from COS modified with Fenton’s reagent for the treatment of cooking fumes was first reported.” Please replace first for “firstly” or “for the first time”.
  5. Line 63: Please provide more information on this process. “ In a typical experimental procedure, COS was first cleaned and dried”.
  6. Line 66: Was washed, perhaps?
  7. Lines 69-72. Please add more information and rephrase for clarity.
  8. Lines 124- 126. “All unmeasured samples were treated for 4 h at 120 °C. The specific surface areas and pore size distributions were obtained in accordance with BET and BJH, respectively; the total pore volume was obtained based on the adsorption quantity of N2 when p/p0 = 0.95.” Please add references for these methods.
  9. Please provide more data for section 1.5. Equipments, conditions…
  10. Lines 166 and 181. Please add the temperature at which the isotherms were obtained.
  11. Line 168. Add reference for isotherms classification.
  12. Lines 196-197. It should be “absorption” peak and not “adsorption”. Please revise in the text for FTIR analysis.
  13. Table 2. Values for SBET don’t require decimal places, please abbreviate.
  14. Figure 9. I would recommend restyling the figure, it’s confusing.
  15. A comparison with the performance of similar materials on the adsorption of cooking fumes would be advisable.
  16. Pay attention to the units. For example, “90 mg/m3, …. 4300 h−1”
  17. I easily found a paper on the mesoporous shell of camellia oleifera, please consider adding this reference.

Author Response

  1. I would advise to reformulate the title. I would suggest something like: “Modified camellia oleifera shell carbon with enhanced performance for the adsorption of cooking fumes”, or similar. It would be more informative and clearer.

Response: Thank you very much for this constructive comment. We have modified the title in the reversed manuscript as you advise.

  1. Lines 39-41: “However, AC has become a kind of scarce resource because some solid wastes such as straws [12, 13], nut shells [14-17], almond shells [18-21], and bagasse [22, 23] are increasingly becoming utilized to prepare AC [1]”. I don’t quite understand this sentence, if the biomasses are increasingly being used, AC would not be a scarce resource…

Response: Thank you very much for this constructive comment. We have modified the sentence as "Recently, AC has become a popular material because many solid wastes such as straws [12, 13], nut shells [14-17], almond shells [18-21], and bagasse [22, 23] can be utilized to prepare AC [1].", which make it understandable and reasonable.   

3.Line 54. Please, replace “aslo” for “also”.

Response: We are sorry for this carelessness. We have replace “aslo” for “also”.

  1. Lines 60-61. “In this research, an AC from COS modified with Fenton’s reagent for the treatment of cooking fumes was first reported.” Please replace first for “firstly” or “for the first time”.

Response: Thank you for your advice. We have replaced "first" for “firstly”

  1. Line 63: Please provide more information on this process. “ In a typical experimental procedure, COS was first cleaned and dried”.

Response: Thank you very much for this constructive comment. We have provided more information on the process of COS pretreatment in the revised manuscript.

  1. Line 66: Was washed, perhaps?

Response: Thank you for your advice. We have replaced "clean" for “washed”

  1. Lines 69-72. Please add more information and rephrase for clarity.

Response: Thank you very much for this constructive comment. We have provided more information and rephrase for clarity in the revised manuscript.

  1. Lines 124- 126. “All unmeasured samples were treated for 4 h at 120 °C. The specific surface areas and pore size distributions were obtained in accordance with BET and BJH, respectively; the total pore volume was obtained based on the adsorption quantity of N2 when p/p0 = 0.95.” Please add references for these methods.

Response: Thank you very much for this constructive comment. We have rephrased in this section for clarity and added a reference[28] for these methods.

  1. Please provide more data for section 1.5. Equipments, conditions…

Response: Thank you very much for this constructive comment. We have provided more information and rephrase for clarity in the revised manuscript.

  1. Lines 166 and 181. Please add the temperature at which the isotherms were obtained.

Response: Thank you very much for this constructive comment. We have added the temperature at which the isotherms were obtained.

  1. Line 168. Add reference for isotherms classification.

Response: Thank you very much for this constructive comment. We have added a reference[30] for the isotherms classification.

  1. Lines 196-197. It should be “absorption” peak and not “adsorption”. Please revise in the text for FTIR analysis.

Response: Thank you for your advice. We have replaced "adsorption" for “absorption” in the text for FTIR analysis.

  1. Table 2. Values for SBET don’t require decimal places, please abbreviate.

Response: Thank you very much for this constructive comment. We have abbreviated

the decimal places of values for SBET in table 2.

  1. Figure 9. I would recommend restyling the figure, it’s confusing.

Response: Thank you very much for this constructive comment. We have deleted Figure 9 to avoid confusion.

  1. A comparison with the performance of similar materials on the adsorption of cooking fumes would be advisable.

Response: Thank you very much for this constructive comment. We have added a table (table 3) and made a comparison with the performance of similar materials on the adsorption of cooking fumes in the revised manuscript.

  1. Pay attention to the units. For example, “90 mg/m3, …. 4300 h−1”

Response: Thank you very much for this constructive comment. We have unified the units.

  1. I easily found a paper on the mesoporous shell of camellia oleifera, please consider adding this reference.

Response: Thank you for your advice. We have added a reference [31].

Reviewer 2 Report

In this work the authors describe the preparation of an adsorbent material, produced by modification of Camellia oleifera shell using phosphoric acid and Fenton’s reagents. The materials were well characterized and tested for the adsorption of cooking fume pollutants, showing promising results. The authors report that, even though COSC-1 is less porous and shows a smaller surface area than COSC-0, it adsorbs more because of the larger number of carbonyl and carboxyl functional groups.

The manuscript could be published after a moderate revision and a general check of the English language.

I suggest to make some changes, as listed below.

Detailed comments:

-Abstract: COSC-0 and COSC-1 should be defined somehow before being used for the first time.

-Section 1.3: volume velocity should be measured as a volume/time. I guess “4300 h-1” is the Gas Hour Space Velocity (GHSV), as indicated in Fig.2. Please, correct the sentence.

- Section 1.5: (line 120) The sentence “The SEM images of samples were acquired by SEM” is quite obvious. It can be replaced by something like “The morphology of the samples was investigated by scanning electron microscopy (SEM)” or “A morphological characterization of the samples was performed by scanning electron microscopy (SEM)”.

(line 127) please define “p/p0

-Fig.2: The legenda is wrong

-Fig.3 and Fig.4: “COA” should be “COS”

-Fig.6: Please, again check the legenda

-Page 8, lines 202-203: “C=O in the hydroxyl” should be “C=O in the carbonyl and carboxyl”;

line 208: I would not write “Figures 8(a) and 8(b)”, because the figure is only one (Figure 8) and (a), (b) identify the spectra in the figure.

-In fig.8 the OH peak at 1620 cm-1 is larger for COSC-0 with respect to the COSC-1 sample. Why? Is it because of a larger amount of adsorbed water? Please, comment on that.

In COSC-1 sample the peak at 592 cm-1 is associated to Fe2+ and Fe3+ due to the Fenton process. Please add a reference. In the same position there is a larger peak also in the sample COSC-0: what is it related to?

Furthermore, you suppose that Fe3+, Fe2+ and H2O2 present on the COSC-1 surface after the Fenton process could decompose pollutants adsorbed on the surface. Could you provide an estimation of Fe amounts present on the surface of COSC-1? Is it possible for the sample COSC-1 to discriminate between the two effects due to the adsorption phenomenon and the degradation induced by Fe ions?

Furthermore, is the presence of H2O2 on the surface necessary for pollutants decomposition by Fe ions to occur? Please, comment on all these issues.

Author Response

  1. -Abstract: COSC-0 and COSC-1 should be defined somehow before being used for the first time.

 Response: Thank you very much for this constructive comment. In the abstract, COSC-0 and COSC-1 have been defined before being used for the first time.

  1. -Section 1.3: volume velocity should be measured as a volume/time. I guess “4300 h-1” is the Gas Hour Space Velocity (GHSV), as indicated in Fig.2. Please, correct the sentence.

 Response: Thank you for your advice. We have corrected the sentence.

  1. - Section 1.5: (line 120) The sentence “The SEM images of samples were acquired by SEM” is quite obvious. It can be replaced by something like “The morphology of the samples was investigated by scanning electron microscopy (SEM)” or “A morphological characterization of the samples was performed by scanning electron microscopy (SEM)”.

 Response: Thank you very much for this constructive comment. The sentence “The SEM images of samples were acquired by SEM” has been replaced by “The morphology of the samples was investigated by scanning electron microscopy (SEM)”

  1. (line 127) please define “p/p0”

Response: Thank you for your advice. The “p/p0” has been defined.

  1.  -Fig.2: The legenda is wrong

Response: Thank you for your advice. The legenda has been corrected.

  1. -Fig.3 and Fig.4: “COA” should be “COS”

Response: Thank you very much for this constructive comment. “COA” has been replaced for “COS” in Fig.3 and Fig.4.

  1. -Fig.6: Please, again check the legenda.

Response: Thank you very much for this constructive comment. The legenda has been corrected.

  1. -Page 8, lines 202-203: “C=O in the hydroxyl” should be “C=O in the carbonyl and carboxyl”;

Response: Thank you very much for this constructive comment. “C=O in the hydroxyl” has been replaced for “C=O in the carbonyl and carboxyl”.

  1. line 208: I would not write “Figures 8(a) and 8(b)”, because the figure is only one (Figure 8) and (a), (b) identify the spectra in the figure.

Response: Thank you very much for this constructive comment. “Figures 8(a) and 8(b)” has been replaced for “Figures 8”.

  1.  -In fig.8 the OH peak at 1620 cm-1 is larger for COSC-0 with respect to the COSC-1 sample. Why? Is it because of a larger amount of adsorbed water? Please, comment on that.

Response: Thank you very much for this constructive comment. In fig.8, the OH peak at 1620 cm-1 is larger for COSC-0 with respect to the COSC-1 sample. We consider it is because of a larger amount of adsorbed water. Comparatively, a new peak appears around 1714 cm−1 indicates the generation of carbonyl and carboxyl groups on the surface of COSC-1 after COSC-0 was modified by Fenton’s reagent, the polarity decrease of COSC-1 leads to the decrease of the adsorbed water of COSC-0.

  1. In COSC-1 sample the peak at 592 cm-1 is associated to Fe2+ and Fe3+ due to the Fenton process. Please add a reference. In the same position there is a larger peak also in the sample COSC-0: what is it related to?

Response: Thank you very much for this constructive comment. According to the previous report, the peak at 500-600 cm-1 belonged to the lattice vibration of metal cations, so we think that the peak at 592 cm-1 is associated to Fe2+ and Fe3+ due to the Fenton process, and we have added a reference [36]. In the same position there is a larger peak also in the sample COSC-0, which is related to the different cations derived from the raw material of COS, after modified, these cations can be resolved in acid medium and removed in the process of washing.

  1. Furthermore, you suppose that Fe3+, Fe2+ and H2O2 present on the COSC-1 surface after the Fenton process could decompose pollutants adsorbed on the surface. Could you provide an estimation of Fe amounts present on the surface of COSC-1? Is it possible for the sample COSC-1 to discriminate between the two effects due to the adsorption phenomenon and the degradation induced by Fe ions?

Response: Thank you very much for this constructive comment. However, this involves the balance between adsorption and catalytic oxidation of cosc-1, which is complicated and needs further study.

  1. Furthermore, is the presence of H2O2 on the surface necessary for pollutants decomposition by Fe ions to occur? Please, comment on all these issues.

Response: Thank you very much for this constructive comment. However, this involves the mechanism of solid phase catalytic reaction, which is complicated and needs further study.

Round 2

Reviewer 1 Report

I recommend the publication of this manuscript in its present form.